# On Learnability and Disambiguation of Multiclass Partial Concept Classes

**Jingyuan Xu** [1]  **Xin Zou** [1]  **Xiuwen Gong** [1]  **Weiwei Liu** [2]

## Abstract

We study the Probably Approximately Correct (PAC) learnability of partial concept classes in the multiclass setting, where the label space can be infinite. While the Natarajan dimension characterizes learnability for finite label spaces, we show it fails when the label space is unbounded. Instead, we prove that the Daniely-Shalev (DS) dimension provides a characterization of learnability for partial concept classes in the general multiclass setting. Furthermore, our analysis reveals a surprising phenomenon we call the "Disambiguation Paradox": disambiguation schemes with simple label space can destroy learnability, while richer labeling may preserve it. We further characterize how the number and structure of disambiguation labels control the induced DS-dimension, yielding a trade-off between label complexity and sample complexity.

## 1. Introduction

In practical learning problems, data usually satisfy specific properties such as a low-dimensional structure (Mardia et al., 1979; Roweis & Saul, 2000; Baraniuk et al., 2010). Traditional PAC setting relies on additional modeling assumptions instead of data itself (Shawe-Taylor et al., 1998; Vovk et al., 2005). Recently, Alon et al. (2021b) have addressed this by introducing *partial concept classes*, which allow abstention on some inputs and naturally model selective prediction. Later Kalavasis et al. (2022) have studied the PAC learnability for multiclass partial concept classes and shown that, when the label space $\mathcal{Y}$ is finite, learnability is characterized by the Natarajan dimension. Their upper bounds, however, scale as $\log|\mathcal{Y}|$ and therefore do not yield guarantees for unbounded label spaces. Inspired by a recent breakthrough in characterization of multiclass total

(non-partial) learning via Daniely–Shalev (DS) dimension (Brukhim et al., 2022; Daniely & Shalev-Shwartz, 2014), a natural question arises:

---
**Question 1**

*Can the DS-dimension fully characterize the PAC learnability of multiclass partial concept classes?*

---

We answer this question in the affirmative: we prove that finiteness of the DS-dimension is both necessary and sufficient for PAC learnability of multiclass partial concept classes, establishing it as the fundamental complexity measure in this regime.

The proof of sufficiency is constructive, highlighting an algorithmic subtlety specific to partial learning: how should a learner handle inputs where the target concept is undefined? Any predictor outputting a definite label on such inputs effectively performs a disambiguation—extending the partial concept to a total one. While Empirical Risk Minimization (ERM) is a common approach in traditional binary learning (Vapnik, 1995; Shalev-Shwartz & Ben-David, 2014), it fails in learning partial concept classes, even in binary setting (Alon et al., 2021b). We argue that ERM is essentially an implicit disambiguation: by selecting a specific hypothesis consistent with the training data, an ERM learner arbitrarily commits to a particular labeling on the undefined regions. As a result, this implicit choice can be catastrophic. This motivates us to study the disambiguation in multiclass setting explicitly.

When designing disambiguation schemes, one might naturally appeal to Occam's Razor: restricting the disambiguation with the minimal necessary set of labels should yield the simplest hypothesis class and thus facilitate learning. **In this work, we challenge this intuition.** Specifically, we investigate whether such "label parsimony" is indeed beneficial in the multiclass setting:

---
**Question 2**

*Does minimizing the number of disambiguation labels necessarily make the learning task easier?*

---

Surprisingly, we answer this question in the negative, revealing a counter-intuitive phenomenon we term the "Disambiguation Paradox". We show that the choice of disambiguation labels can have a decisive effect on the induced

[1]School of Computer Science, Wuhan University, China. [2]Institute of Big Data, Fudan University, China. Correspondence to: Xiuwen Gong <gongxiuwen@gmail.com>, Weiwei Liu <liuweiwei863@gmail.com>.

*Proceedings of the 43rd International Conference on Machine Learning*, Seoul, South Korea. PMLR 306, 2026. Copyright 2026 by the author(s).

DS-dimension: there are learnable partial classes for which any binary disambiguation yields a total class of infinite DS-dimension, while a hypothesis-specific disambiguation over a larger label set preserves the DS-dimension. These results motivate a quantitative trade-off between *sample complexity* and the *label complexity* required by disambiguation schemes.

**Conflict of Interest Disclosure.** The authors declare no financial conflicts of interest or other substantive conflicts of interest that could reasonably be perceived to influence this work.

## 1.1. Related Works

**PAC learning theory.** Probably Approximately Correct (PAC) Learning is the foundation of modern learning theory (Valiant, 1984; Vapnik, 1995), with binary learnability controlled by uniform convergence (Vapnik & Chervonenkis, 1971) and characterized by the VC dimension (Blumer et al., 1989). The framework has been extended to the multiclass setting (Natarajan, 1989; Ben-David et al., 1995; Daniely et al., 2011; Daniely & Shalev-Shwartz, 2014; Brukhim et al., 2022), where Daniely et al. (2011) and Daniely & Shalev-Shwartz (2014) identify ERM gaps and motivate new dimensions beyond the Natarajan dimension (Natarajan, 1989), culminating in the DS-dimension characterization of Brukhim et al. (2022). The DS-dimension is further leveraged for multiclass boosting by Brukhim et al. (2023), and Zou et al. (2024) establish convergence for AdaBoost.MH with factorized multi-class classifiers. For regression, scale-sensitive dimensions characterize learnability (Alon et al., 1997; Bartlett & Long, 1998; Simon, 1997), with Attias et al. (2023) obtaining optimal learners in both PAC and online realizable regression. For online learning (Littlestone, 1987; Ben-David et al., 2009), Alon et al. (2021a) establish optimal regret via adversarial laws of large numbers. The theory has also been extended to robust learning (Montasser et al., 2019; Xu & Liu, 2022; 2023; Montasser et al., 2022; Attias et al., 2022b;a).

**Partial concept classes.** Partial concept classes are formalized by Alon et al. (2021b), who develop a PAC learning theory based on the VC dimension; an earlier study of $\{0, *, 1\}$-valued hypotheses appears in Long (2001), and partial concepts also arise in comparative learning (Hu & Peale, 2023). A closely related but distinct line of work studies *partial label learning* and *partial multi-label learning*, where each instance carries a candidate label set rather than abstention regions (Gong et al., 2021; 2022b;a; 2023). Cheung et al. (2023) studies online learnability and disambiguation of partial concept classes, and Pabbaraju (2024) uses disambiguation to prove that multiclass learnability does not imply sample compression, only in the context of total learning. Recently, Kalavasis et al. (2022) have

shown that for finite label spaces, partial concept classes are learnable whenever the Natarajan dimension is finite; their characterization, however, crucially relies on the finiteness of the label space and does not extend to general settings. This work fills that gap and develops a theory of partial concept classes under general multiclass disambiguation.

## 2. Preliminaries

In the remainder of this article, we denote the set $\{1, \ldots, n\}$ (for $n \in \mathbb{N}$) by $[n]$. If $A$ and $B$ are sets, we use $B^A$ to denote the collection of all mappings from $A$ to $B$ and $2^A$ to denote the power set of $A$, that is the collection of all subsets of $A$. Let $C = \{c_1, \ldots, c_m\} \subseteq \mathcal{X}$ and $C' \subseteq C$ be a subset of $C$, then $I_{C'} \subseteq [m]$ denotes the index set of $C'$. Let $\mathcal{H}$ be a class of functions defined in $\mathcal{X}$, then $\mathcal{H}|_C$ represents the restriction of $\mathcal{H}$ to $C$, that is $\mathcal{H}_C = \{(h(c_1), \ldots, h(c_m)) : h \in \mathcal{H}\}$. Finally, we denote the indicator function by $\mathbb{1}(\text{event})$, that is $1$ if an event happens and $0$ otherwise.

### 2.1. Partial Concept Classes

Let $\mathcal{X}$ denote an instance space and $\mathcal{Y}$ a label space. In the multiclass setting with possibly infinitely many labels, we allow $|\mathcal{Y}| = \infty$. A *partial concept* (Alon et al., 2021b) $h : \mathcal{X} \to \mathcal{Y} \cup \{\star\}$ is a function that can output a special symbol $\star$, representing "undefined". For any partial concept $h$, its *support* is

$$\text{supp}(h) = \{x \in \mathcal{X} : h(x) \neq \star\}.$$

Intuitively, partial concepts only need to define outputs on subsets of $\mathcal{X}$, and are permitted to abstain elsewhere. This captures settings where the meaningful portion of the domain (e.g., a region with margin or a structured manifold) is a proper subset of $\mathcal{X}$. Additionally, if a function satisfies $\text{supp}(h) = \mathcal{X}$, we call it a *total concept*. In the remainder of this paper, we use calligraphic font (e.g., $\mathcal{H}$) or overlined blackboard bold (e.g., $\bar{\mathbb{H}}$) to denote a total concept class (or hypothesis class)[1], and blackboard bold (e.g., $\mathbb{H}$) to denote a partial concept class.

A center problem in partial concept classes learning is if one can find a total class that can represent the original partial one. This idea introduces the definition of *disambiguation*, describing how a total class represents a partial class. (Alon et al., 2021b):

**Definition 2.1** (Disambiguation scheme)**.** Let $\mathbb{H} \subseteq (\mathcal{Y} \cup \{\star\})^{\mathcal{X}}$ be a partial concept class. A disambiguation scheme is a mapping $\varphi : (\mathcal{Y} \cup \{\star\})^{\mathcal{X}} \to \mathcal{Y}^{\mathcal{X}}$ s.t.

$$\varphi(h)(x) = h(x) \quad \text{for all } x \in \text{supp}(h).$$

---

[1]The terms *total concept* and *hypothesis* refer to the same object; we use them interchangeably.

We say that a total concept class $\bar{\mathbb{H}}_\varphi$ *strongly disambiguates* $\mathbb{H}$ via $\varphi$ if

$$\bar{\mathbb{H}}_\varphi = \{\varphi(h) : h \in \mathbb{H}\}.$$

Equivalently, for every $h \in \mathbb{H}$ there exists $\bar{h} \in \bar{\mathbb{H}}_\varphi$ such that $\bar{h}(x) = h(x)$ for all $x \in \text{supp}(h)$.

Intuitively, disambiguation resolves the ambiguity of the undefined symbol $\star$ by specifying how to label points outside the support of a partial concept.

## 2.2. PAC Learning Problem

Under the framework of partial concept classes, the learning problem is defined as follows. Fix an unknown distribution $\mathcal{D}$ over $\mathcal{X} \times \mathcal{Y}$ and a partial concept class $\mathbb{H} \subseteq (\mathcal{Y} \cup \{\star\})^\mathcal{X}$. The goal of a learner is to output a function $h$ (not necessarily in $\mathbb{H}$) based on $n$ labeled training samples $S$ drawn i.i.d. from $\mathcal{D}$, such that the *risk* of $h$ under $\mathcal{D}$

$$R_\mathcal{D}(h) \triangleq \underset{(x,y)\sim\mathcal{D}}{\mathbb{E}} [l(h(x), y)]$$

is minimal for the 0-1 loss $l(\hat{y}, y) \triangleq \mathbb{1}(\hat{y} \neq y)$, and abstentions $h(x) = \star$ are treated as errors, i.e. $l(\star, y) = l(y, \star) = 1$ for all $y \in \mathcal{Y}$.

We study the learning problem from the PAC viewpoint and begin by working under the *realizability assumption* for partial concept classes.: a distribution $\mathcal{D}$ is said to be realizable by $\mathbb{H}$ if, with probability 1 over the draw of the sample $S \sim \mathcal{D}^n$, there exists an $h \in \mathbb{H}$ such that

$$\{x_i\}_{i=1}^n \subseteq \text{supp}(h), \qquad h(x_i) = y_i \text{ for all } i \leq n.$$

In other words, all observed samples lie in the support of some partial hypothesis that labels them correctly.

**Definition 2.2** (Realizable PAC learnability). A partial concept class $\mathbb{H} \subseteq (\mathcal{Y} \cup \{\star\})^\mathcal{X}$ is PAC learnable if, for every $\varepsilon, \delta \in (0,1)$, there exists a finite $\mathcal{M}_{re} : (0,1)^2 \to \mathbb{N}$ and a learning algorithm $\mathcal{A}$ such that, for every distribution $\mathcal{D}$ on $\mathcal{X} \times \mathcal{Y}$ realizable w.r.t. $\mathbb{H}$, for $S \sim \mathcal{D}^{\mathcal{M}_{re}(\varepsilon,\delta)}$, with probability at least $1 - \delta$,

$$R_\mathcal{D}(\mathcal{A}(S)) \leq \varepsilon.$$

The mapping $\mathcal{M}_{re}(\varepsilon, \delta)$ is called the sample complexity of $\mathcal{A}$.

The realizable assumption is equivalent to the *approximation error* of $\mathbb{H}$ is 0, i.e., $R_\mathcal{D}^*(\mathbb{H}) := \min_{h \in \mathbb{H}} R_\mathcal{D}(h) = 0$. Generally, we will be interested in achieving prediction error not-much-worse than $R_\mathcal{D}^*(\mathbb{H})$, introducing the agnostic PAC setting:

**Definition 2.3** (Agnostic PAC learnability). A partial concept class $\mathbb{H} \subseteq (\mathcal{Y} \cup \{\star\})^\mathcal{X}$ is agnostically PAC learnable if, for every $\varepsilon, \delta \in (0,1)$, there exists a finite $\mathcal{M}_{ag} : (0,1)^2 \to$ $\mathbb{N}$ and a learning algorithm $\mathcal{A}$ such that, for every distribution $\mathcal{D}$ on $\mathcal{X} \times \mathcal{Y}$, for $S \sim \mathcal{D}^{\mathcal{M}_{ag}(\varepsilon,\delta)}$, with probability at least $1 - \delta$,

$$R_\mathcal{D}(\mathcal{A}(S)) \leq R_\mathcal{D}^*(\mathbb{H}) + \varepsilon.$$

The mapping $\mathcal{M}_{ag}(\varepsilon, \delta)$ is known as the sample complexity of $\mathcal{A}$ for agnostic PAC learning.

We then recall some known results regarding the sample complexity of learning partial concept classes. Recall the definition of the Vapnik-Chervonenkis dimension(Vapnik & Chervonenkis, 1971; Vapnik, 1995).

**Definition 2.4** (VC-dimension). Let $\mathcal{H} \subseteq \{0,1\}^\mathcal{X}$ be a hypothesis class. A subset $F \subseteq \mathcal{X}$ is shattered by $\mathcal{H}$ if $\mathcal{H}|_F = \{0,1\}^F$. The VC-dimension of $\mathcal{H}$, denoted $\text{VC}(\mathcal{H})$, is the maximal cardinality of a subset $F \subseteq \mathcal{X}$ that is shattered by $\mathcal{H}$.

The VC-dimension characterizes the sample complexity of learning binary (partial) classes, as the following bounds suggests.

**Theorem 2.5** (Alon et al., 2021b). *For any partial concept class $\mathbb{H}$ with $\text{VC}(\mathbb{H})$, the sample complexity $\mathcal{M}_{re}(\varepsilon, \delta)$ of PAC learning $\mathbb{H}$ satisfying:*

$$\mathcal{M}_{re}(\varepsilon, \delta) = O\left(\frac{\text{VC}(\mathbb{H})}{\varepsilon} \log^2\left(\frac{\text{VC}(\mathbb{H})}{\varepsilon}\right) + \frac{1}{\varepsilon} \log\left(\frac{1}{\delta}\right)\right),$$

$$\mathcal{M}_{re}(\varepsilon, \delta) = \Omega\left(\frac{\text{VC}(\mathbb{H})}{\varepsilon} + \frac{1}{\varepsilon} \log\left(\frac{1}{\delta}\right)\right).$$

*Moreover, if $\text{VC}(\mathbb{H}) = \infty$, then $\mathbb{H}$ is not PAC learnable.*

A fundamental difference between learning a total class and a partial class is that the empirical risk minimization (ERM) principle can fail for partial concept classes (Alon et al., 2021b).

It is natural to seek a generalization of the VC-dimension to hypothesis classes of non-binary functions. In classical learning theory, two natural generalizations of the VC dimension to the multiclass setting are the graph dimension and the Natarajan dimension, both introduced by Natarajan (1989). These notions refine the definition of shattering of a set to multiclass setting. In this work, we only introduce the Natarajan dimension.

**Definition 2.6** (Natarajan dimension). Let $\mathcal{H} \subseteq \mathcal{Y}^\mathcal{X}$ be a hypothesis class. A subset $F \subseteq \mathcal{X}$ is N-shattered by $\mathcal{H}$ if there exists $f_1, f_2 : F \to \mathcal{Y}$ such that $\forall x \in F, f_1(x) \neq f_2(x)$, and for every $G \subseteq F$, there is a $g \in \mathcal{H}$ such that

$$\forall x \in G, g(x) = f_1(x), \quad \text{and} \quad \forall x \in F \backslash G, g(x) = f_2(x).$$

The Natarajan dimension of $\mathcal{H}$, denoted by $\text{Ndim}(\mathcal{H})$, is the maximal cardinality of a set that is N-shattered by $\mathcal{H}$.

The Natarajan dimension has been shown to characterize PAC learnability in the standard multiclass learning setting when the label space is finite. This result is then extended to partial learning:

**Theorem 2.7** (Kalavasis et al., 2022). *For any partial class* $\mathbb{H} \subseteq \{0, \dots, k, \star\}^{\mathcal{X}}$ *with* $\mathrm{Ndim}(\mathbb{H}) < \infty$, *the sample complexity* $\mathcal{M}_{re}(\varepsilon, \delta)$ *of PAC learning* $\mathbb{H}$ *satisfying:*

$$\mathcal{M}_{re}(\varepsilon, \delta) = O\left(\frac{\mathrm{Ndim}(\mathbb{H}) \log(k)}{\varepsilon} \log\left(\frac{1}{\delta}\right)\right),$$

$$\mathcal{M}_{re}(\varepsilon, \delta) = \Omega\left(\frac{\mathrm{Ndim}(\mathbb{H})}{\varepsilon} + \frac{1}{\varepsilon} \log\left(\frac{1}{\delta}\right)\right).$$

*Moreover, if* $\mathrm{Ndim}(\mathbb{H}) = \infty$, *then* $\mathbb{H}$ *is not PAC learnable.*

However, the upper bound in Theorem 2.7 depends on $\log(k)$ and therefore does not apply when $k = |\mathcal{Y}| = \infty$. Moreover, for total multiclass classes with unbounded label spaces, finite Natarajan dimension does not characterize PAC learnability (Brukhim et al., 2022). In that setting, the appropriate parameter is the Daniely–Shalev (DS) dimension (Daniely & Shalev-Shwartz, 2014), which was shown by Brukhim et al. (2022) to fully characterize multiclass PAC learnability of total concept classes and was subsequently used by Brukhim et al. (2023) to obtain multiclass boosting guarantees independent of the number of classes. We conclude this section by recalling the definition of DS-dimension and the corresponding characterization theorem.

**Definition 2.8** (Pseudo-cube). *A hypothesis class* $\mathcal{H} \in \mathcal{Y}^d$ *is called a* pseudo-cube *of dimension* $d$ *if it is non-empty, finite and for every* $h \in \mathcal{H}$ *and* $i \in [d]$, *there exists* $g \in \mathcal{H}$ *such that* $g(i) \neq h(i)$ *and* $g(j) = h(j)$ *for all* $j \neq i$.

**Definition 2.9** (DS-dimension (Daniely & Shalev-Shwartz, 2014)). *Let* $\mathcal{H} \subseteq \mathcal{Y}^{\mathcal{X}}$ *be a hypothesis class. A subset* $F \in \mathcal{X}^n$ *is DS-shattered by* $\mathcal{H}$ *if* $\mathcal{H}|_F$ *contains a* $n$-*dimensional pseudo-cube. The DS-dimension of* $\mathcal{H}$, *denoted by* $\mathrm{DS}(\mathcal{H})$ *is the maximal cardinality of a set that is DS-shattered by* $\mathcal{H}$.

**Theorem 2.10** (Brukhim et al., 2022). *A hypothesis class* $\mathcal{H} \subseteq \mathcal{Y}^{\mathcal{X}}$ *is (agnostically) PAC learnable if and only if the DS-dimension of* $\mathcal{H}$ *is finite, i.e.,* $\mathrm{DS}(\mathcal{H}) < \infty$.

## 3. Motivating Phenomenon

In this section, we show that multiclass partial learning is fundamentally different from classical learning, from both the characterization and algorithmic perspectives.

### 3.1. Natarajan Dimension Fails to Characterize Multiclass Partial Learnability

We start by showing that once the finiteness assumption on the label space ($|\mathcal{Y}| < \infty$) is removed, the Natarajan

dimension is no longer sufficient to characterize learnability of multiclass partial concept classes.

In particular, there exists a partial class with Natarajan dimension 1 which is nevertheless not PAC learnable.

**Theorem 3.1** (Finite Natarajan dimension does not imply learnability). *There exists a partial concept class* $\mathbb{H} \subseteq (\mathcal{Y} \cup \{\star\})^{\mathcal{X}}$ *such that*

- *its Natarajan dimension satisfies* $\mathrm{Ndim}(\mathbb{H}) = 1$;

- $\mathbb{H}$ *is not PAC learnable, even in the realizable case.*

The proof is presented in Appendix B

### 3.2. Implicit Disambiguation and the Failure of ERM

In the context of partial learning, a fundamental challenge is how to handle inputs outside of the support of a hypothesis, where the label is $\star$. Any total predictor that outputs a specific label in $\mathcal{Y}$ for every $x \in \mathcal{X}$, effectively makes a choice about how to fill in these missing values.

While the Fundamental Theorem of PAC Learning guarantees that Empirical Risk Minimization (ERM) works for total binary classes with finite VC-dimension (Vapnik, 1995; Shalev-Shwartz & Ben-David, 2014), ERM is known to be insufficient in more general settings: in the multiclass setting, gaps between ERM learnability and PAC learnability were established by Daniely et al. (2011) and Daniely & Shalev-Shwartz (2014), and further developed by Brukhim et al. (2022) via the DS dimension; for partial concept classes, Alon et al. (2021b) exhibit binary partial classes that are PAC learnable but for which ERM fails.

At a high level, the difficulty of learning partial concept classes is that any total learner must assign labels outside the support of a partial hypothesis, thereby implicitly choosing a disambiguation. The following theorem gives an explicit example where a generic ERM learner fails even though the underlying partial class is PAC learnable.

**Theorem 3.2** (Failure of ERM (Alon et al., 2021b)). *There exists a partial concept class* $\mathbb{H} \subseteq \{0, 1, \star\}^{\mathcal{X}}$ *such that:*

- $\mathrm{VC}(\mathbb{H}) = 1$, *hence* $\mathbb{H}$ *is PAC learnable.*

- *For any total concept class* $\bar{\mathbb{H}}$, *there exists an ERM algorithm for* $\bar{\mathbb{H}}$ *that fails to PAC learn* $\mathbb{H}$.

**ERM as implicit disambiguation.** An ERM procedure that is consistent with the training data necessarily commits—through its tie-breaking rule and inductive bias—to a particular completion on the unobserved region. Without explicit control, this completion may induce a total class whose DS-dimension is much larger than that of the underlying partial class (and can even be infinite), thereby destroying

learnability; see Section 5 (Theorem 5.3) for an explicit manifestation of this effect. This perspective clarifies why ERM alone is insufficient in the partial setting and motivates algorithms that explicitly regulate off-support behavior.

This result also highlights that the algorithmic landscape is more complex than in the classical setting, often requiring specialized algorithms rather than simple risk minimization. In the next section we utilize the sample compression scheme to establish learners for partial DS-classes.

# 4. DS-dimension as a Characterization of Multiclass Partial Learning

In this section we establish sample-complexity bounds in terms of the DS-dimension. As a result, finiteness of the DS-dimension is a necessary and sufficient condition for learning multiclass partial classes.

## 4.1. Upper Bounds of Sample Complexity

We start from the realizable case and upper bound the sample complexity of learning a partial concept class in the multiclass setting. It is well known that a compression scheme implies learnability (Littlestone & Warmuth, 1986; David et al., 2016). The sample compression scheme is formally defined as follows.

**Definition 4.1** (Sample Compression Scheme). A sample compression scheme for $\mathcal{H} \subseteq \mathcal{Y}^{\mathcal{X}}$ with kernel size $r$ is a pair $(\kappa, \rho)$, consisting of a compression function $\kappa : (\mathcal{X} \times \mathcal{Y})^* \to (\mathcal{X} \times \mathcal{Y})^r$ and a reconstruction function $\rho : (\mathcal{X} \times \mathcal{Y})^r \to \mathcal{Y}^{\mathcal{X}}$, satisfying that for any $m \geq r$ and $S = \{(x_i, y_i)\}_{i=1}^m \in (\mathcal{X} \times \mathcal{Y})^m$ that is realizable w.r.t. $\mathcal{H}$, $\rho(\kappa(S))(x_i) = y_i$ for all $i \in [m]$.

This definition can be naturally adapted to learning partial concept classes. Our approach to constructing the compression algorithm is broadly inspired by (Brukhim et al., 2022) and proceeds in the following steps:

1. obtain a weak transductive guarantee using the *one-inclusion graph (OIG)* algorithm;

2. control the uncertainty induced by an infinite label space via a *menu*;

3. combine two compression components to yield an *improper* learner.

We now formalize the required tools.

**The One-Inclusion Graph (OIG) algorithm.** To derive a compression scheme for a partial concept class, we use the *one-inclusion graph* algorithm (Haussler et al., 1994; Rubinstein et al., 2006). The formal definition involves graph theory and is deferred to Appendix A.1. Informally, OIG

provides a transductive predictor: given $n$ unlabeled examples $S_{\mathcal{X}} \in \mathcal{X}^n$, one point is drawn uniformly at random, and the algorithm must predict its label given the labels of the remaining $n - 1$ points.

A key ingredient is the following guarantee for the one-inclusion graph algorithm, showing that finite DS-dimension yields a non-trivial transductive predictor for $\mathbb{H}$.

**Theorem 4.2.** *Let $\mathbb{H}$ be a partial concept class with $d = \mathrm{DS}(\mathbb{H}) < \infty$. Then the one-inclusion algorithm $\mathcal{A}_{\mathbb{H}}^{\mathrm{OIG}}$ (see Algorithm 2 in Appendix A.1) satisfies that: For every $\mathbb{H}$-realizable sample $S' = \{(x'_1, y'_1), \ldots, (x'_{d+1}, y'_{d+1})\}$, there exists an $i \in [d+1]$ such that $h_{S'_{-i}}(x'_i) = y'_i$, where $S'_{-i}$ is the set after removing the $i$-th example from $S'$ and $h_{S'_{-i}} = \mathcal{A}_{\mathbb{H}}^{\mathrm{OIG}}(S'_{-i})$.*

The proof is in Appendix A.2.

Following Brukhim et al. (2022), in the multiclass learning setting where the label space may be infinitely large, the constructed sample compression scheme of a DS-partial concept class has two main components, presented in Theorem 4.5 and Theorem 4.9, respectively.

**Component I: list compression.** The first component is a list-compression step in the list-learning framework (Brukhim et al., 2022; Charikar & Pabbaraju, 2023): at test time, a list predictor outputs a *set* of candidate labels rather than a single label, formalized via a *p-menu* and a *list sample compression scheme*.

**Definition 4.3** (*p*-menu). A menu of size $p \in \mathbb{N}$ is a function $\mu : \mathcal{X} \to \{Y \subseteq \mathcal{Y} : |Y| \leq p\}$.

**Definition 4.4** (List Sample Compression Scheme). An $n \to r$ list sample compression scheme with menu size $p$ consists of a *reconstruction function*

$$\rho : (\mathcal{X} \times \mathcal{Y})^r \to \{Y \subseteq \mathcal{Y} : |Y| \leq p\}^{\mathcal{X}}$$

such that for every $\mathbb{H}$-realizable $S \in (\mathcal{X} \times \mathcal{Y})^n$, there exists $S' \in (\mathcal{X} \times \mathcal{Y})^r$ whose elements appear in $S$ such that for every $(x, y)$ in $S$ we have $y \in \mu(x)$, where $\mu = \rho(S')$.

The existence of the first component of compression is guaranteed by the following theorem:

**Theorem 4.5.** *Let $\mathbb{H} \subseteq (\mathcal{Y} \cup \{\star\})^{\mathcal{X}}$ be a partial concept class with $\mathrm{DS}(\mathbb{H}) < \infty$. For every integers $n, t > 0$, there exists an $n \to r_1$ list sample compression scheme for $\mathbb{H}$ with menu size $p$, where*

$$r_1 \leq \frac{\mathrm{DS}(\mathbb{H}) + t + 1}{t + 1}(\mathrm{DS}(\mathbb{H}) + t)\log(n)$$

*and*

$$p \leq \binom{\mathrm{DS}(\mathbb{H}) + t + 1}{t + 1}\log(n).$$

The proof follows the analogous argument for total classes in Brukhim et al. (2022); we defer details to Appendix A.3.

**Component II: sample compression given a menu.** We now turn to the second component of the compression scheme. The key idea is to use a menu to control the uncertainty caused by an infinite label space: the menu restricts the set of plausible labels at each point to a bounded list, enabling multiclass tools to be applied locally even when $|\mathcal{Y}| = \infty$.

We start with the idealized scenario where a menu completely captures the unknown distribution $\mathcal{D}$, leading to the following definition.

**Definition 4.6** (Menu Realizability). A sample $S \in (\mathcal{X} \times \mathcal{Y})^n$ is *realizable* by the menu $\mu$ if $y \in \mu(x)$ for every $(x, y) \in S$. A distribution $\mathcal{D}$ over $\mathcal{X} \times \mathcal{Y}$ is realizable by $\mu$ if every $S \sim \mathcal{D}^m$ is realizable by $\mu$ almost surely.

---

**Algorithm 1** One-inclusion algorithm $\mathcal{A}_{\mathbb{H},\mu}^{\mathrm{OIG}}$ for a partial concept class $\mathbb{H}$ and menu $\mu$

---

**Input:** A sample $S = \{(x_i, y_i)\}_{i=1}^n$ realizable by $\mathbb{H}$ and $\mu$.
**Output:** A hypothesis $h_S : \mathcal{X} \to \mathcal{Y}$.

1: Define $\mathcal{H} \subseteq \mathcal{Y}^n$ as the class of all patterns on the unlabelled data that are realizable by both $\mathbb{H}$ and $\mu$. That is, the set of all $h \in \mathcal{H}|_{\{x_1,\ldots,x_n\}}$ so that $h(i) \in \mu(x_i)$ for $i \in [n]$.
2: Let $\mathcal{A}_{\mathcal{H}}$ be the one-inclusion graph algorithm for $\mathcal{H}$.
3: Set $h_S = \mathcal{A}_{\mathcal{H}}(S)$.

---

Under the menu realizability assumption, we generalize $\mathcal{A}_{\mathbb{H}}^{\mathrm{OIG}}$ to the case when there is a menu, denoted as $\mathcal{A}_{\mathbb{H},\mu}^{\mathrm{OIG}}$. The detail of $\mathcal{A}_{\mathbb{H},\mu}^{\mathrm{OIG}}$ is presented in Algorithm 1. The performance of $\mathcal{A}_{\mathbb{H},\mu}^{\mathrm{OIG}}$ is controlled by the size of the menu $\mu$ and the Natarajan dimension of $\mathbb{H}$ in transductive learning setting, characterized by the following lemma.

**Lemma 4.7** (Natarajan Classes with Menu Realizability are Learnable). *Let $\mathbb{H} \subseteq (\mathcal{Y} \cup \{\star\})^{\mathcal{X}}$ be a partial concept class with Natarajan dimension $\mathrm{Ndim}(\mathbb{H}) = d_N < \infty$ and let $\mu$ be a p-menu. Then $\mathcal{A}_{\mathbb{H},\mu}^{\mathrm{OIG}}$ (Algorithm 1) satisfies: For every distribution $\mathcal{D}$ over $\mathcal{X} \times \mathcal{Y}$ that is realizable by both $\mathcal{H}$ and $\mu$, and for all integers $n > 0$,*

$$\Pr_{(S,(x,y))\sim\mathcal{D}^{n+1}} \left[ \mathcal{A}_{\mathbb{H},\mu}^{\mathrm{OIG}}(S)(x) \neq y \right] \leq \frac{20 d_N \ln(p)}{n}.$$

The proof is presented in Appendix A.4.

We now remove the realizability assumption to derive the second component of the compression scheme, i.e., a sample compression scheme when there is a menu, which is defined as follows:

**Definition 4.8** (Sample Compression Scheme for a Menu). An $n \to r$ *sample compression scheme* for a class $\mathbb{H}$ and a

menu $\mu$ consists of a *reconstruction function*

$$\rho : (\mathcal{X} \times \mathcal{Y})^r \to \mathcal{Y}^{\mathcal{X}}$$

such that for every $S \in (\mathcal{X} \times \mathcal{Y})^n$ that is realizable by both $\mathbb{H}$ and $\mu$, there exists $S' \in (\mathcal{X} \times \mathcal{Y})^r$ whose elements appear in $S$ such that for every $(x, y)$ in $S$ we have $h(x) = y$, where $h = \rho(S')$.

The following theorem demonstrates that partial concept classes with menu and finite Natarajan dimension are compressible.

**Theorem 4.9.** *Let $\mathbb{H} \subseteq (\mathcal{Y} \cup \{\star\})^{\mathcal{X}}$ be a partial concept class with $\mathrm{Ndim}(\mathbb{H}) < \infty$ and let $\mu$ be a p-menu. For every integer $n > 0$, there exists an $n \to r_2$ sample compression scheme for $\mathbb{H}$ and $\mu$ with*

$$r_2 \leq 640 \mathrm{Ndim}(\mathbb{H}) \log(p) \log(n).$$

The proof follows (Brukhim et al., 2022) with a minor improvement in constants and is deferred to Appendix A.4.

Since $\mathrm{Ndim}(\mathcal{H}) \leq \mathrm{DS}(\mathcal{H})$ for every (partial) hypothesis class $\mathcal{H}$ (Daniely & Shalev-Shwartz, 2014), we conclude that every DS-class admits such a scheme.

**Composition: DS-bounded partial classes are compressible.** Combining Theorem 4.5 and Theorem 4.9 gives the final compression scheme:

**Theorem 4.10** (DS-Partial Concept Classes are Compressible). *Let $\mathbb{H} \subseteq (\mathcal{Y} \cup \{\star\})^{\mathcal{X}}$ be a partial concept class with $\mathrm{DS}(\mathbb{H}) < \infty$ and Natarajan dimension $d_N$. For every integers $n, t > 0$, there exists an $n \to r$ sample compression scheme for $\mathbb{H}$ with*

$$r \leq \left( \frac{\mathrm{DS}(\mathbb{H}) + t + 1}{t + 1}(\mathrm{DS}(\mathbb{H}) + t) \right.$$
$$\left. + 640 d_N \log\left( \binom{\mathrm{DS}(\mathbb{H}) + t + 1}{t + 1} \log(n) \right) \right) \log(n). \tag{1}$$

The proof is in Appendix A.5.

Standard compression-based generalization upper bounds (Littlestone & Warmuth, 1986; David et al., 2016) yield:

**Theorem 4.11.** *For every $\mathcal{H} \subseteq (\mathcal{Y} \cup \{\star\})^{\mathcal{X}}$,*

$$\mathcal{M}_{re}(\varepsilon, \delta) = O\left( \frac{\mathrm{DS}(\mathbb{H})^{\frac{3}{2}}}{\varepsilon} \log^3\left( \frac{\mathrm{DS}(\mathbb{H})^{\frac{3}{2}}}{\varepsilon} \right) + \frac{\log(1/\delta)}{\varepsilon} \right),$$

$$\mathcal{M}_{ag}(\varepsilon, \delta) = O\left( \frac{\mathrm{DS}(\mathbb{H})^{\frac{3}{2}}}{\varepsilon^2} \log^3\left( \frac{\mathrm{DS}(\mathbb{H})^{\frac{3}{2}}}{\varepsilon} \right) + \frac{\log(1/\delta)}{\varepsilon^2} \right).$$

The proof is presented in Appendix C.

**About the** $\log(n)$ **dependence.** Our constructed sample compression scheme has size $O(\log n)$ in the sample size, unlike the $n$-independent compression schemes in binary learning (David et al., 2016). One might wonder if the $\log n$ factor in our construction is an artifact of our proof technique or a fundamental property of multiclass learning. Pabbaraju (2024) shows this is unavoidable: there exist multiclass classes with finite DS-dimension that admit no compression scheme of size independent of $n$, implying an intrinsic gap between binary and multiclass learnability.

### 4.2. Lower Bounds of Sample Complexity

In this section we complement the lower bounds, showing that finiteness of the DS-dimension is also necessary for learnability of multiclass partial concept classes.

**Theorem 4.12.** *For every $\mathcal{H} \subseteq (\mathcal{Y} \cup \{\star\})^{\mathcal{X}}$,*

$$\mathcal{M}_{re}(\varepsilon, \delta) = \Omega\left(\frac{\mathrm{DS}\,(\mathbb{H})}{\varepsilon} + \frac{\log(1/\delta)}{\varepsilon}\right),$$

$$\mathcal{M}_{ag}(\varepsilon, \delta) = \Omega\left(\frac{\mathrm{DS}\,(\mathbb{H})}{\varepsilon} + \frac{\log(1/\delta)}{\varepsilon}\right).$$

The proof adapts the binary VC-based argument of (Alon et al., 2021b) to the DS-dimension (Appendix D).

Combining Theorem 4.11 and Theorem 4.12, the DS-dimension characterizes learnability of partial concept classes in the general multiclass setting.

## 5. Disambiguation Paradox and Label–Sample Complexity Trade-offs

In learning theory, reductions to small label spaces are often used as a modeling convenience. For partial concept classes, a natural reduction is to replace abstentions by explicit labels via disambiguation. In Section 2.1 (Definition 2.1), we introduce disambiguation schemes as a way to map partial concepts to total ones. In this section we will show that, in the multiclass setting, the number of labels used for disambiguation can have a structural effect on learnability through the DS-dimension, leading to a trade-off between label complexity and sample complexity.

### 5.1. Disambiguation Paradox

Recall that a disambiguation scheme $\varphi$ turns each partial hypothesis $h \in \mathbb{H}$ into a total hypothesis $\bar{h}$ that agrees with $h$ on $\mathrm{supp}(h)$.

It is natural to ask whether learning a partial class can be reduced to learning a corresponding total class obtained by disambiguation. In the binary case, Alon et al. (2021b) show that such reductions can be inherently costly: there exists a PAC learnable partial class $\mathbb{H}$ for which *every* disambiguation scheme yields a total class of unbounded VC-dimension,

so that no disambiguated total class is PAC learnable, despite $\mathbb{H}$ itself being learnable.

**Theorem 5.1** (Alon et al., 2021b)**.** *There exists a partial concept class $\mathbb{H} \subseteq \{0, 1, \star\}^{\mathbb{N}}$ with $\mathrm{VC}\,(\mathbb{H}) = 1$ while $|\bar{\mathbb{H}}| \geq n^{\Omega(\log n)}$ for every disambiguation $\bar{\mathbb{H}} \subseteq \{0, 1\}^{\mathbb{N}}$ of $\mathbb{H}$, which implies $\mathrm{VC}\,(\bar{\mathbb{H}}) = \infty$.*

In the multiclass setting, a disambiguation scheme may introduce additional labels (beyond those appearing on supports) to label points outside $\mathrm{supp}(h)$. We will quantify the set of labels used off-support by the following notion.

**Definition 5.2** (Disambiguation label domain)**.** Let $\mathbb{H} \subseteq (\mathcal{Y} \cup \{\star\})^{\mathcal{X}}$ be a partial concept class and $\varphi : (\mathcal{Y} \cup \{\star\})^{\mathcal{X}} \to \mathcal{Y}^{\mathcal{X}}$ be a disambiguation scheme of $\mathbb{H}$. The disambiguation label domain of $\varphi$ is defined as

$$\mathcal{Y}_{\varphi} := \{\varphi(h)(x) : h \in \mathbb{H},\ x \in \mathcal{X} \backslash \mathrm{supp}(h)\}.$$

We say that $\mathbb{H}$ admits a disambiguation using a label set $A \subseteq \mathcal{Y}$ if there exists a disambiguation scheme $\varphi$ such that $\mathcal{Y}_{\varphi} = A$.

The next theorem exhibits the *disambiguation paradox*: allowing a richer set of disambiguation labels can preserve learnability (via finite DS-dimension), whereas restricting to a very small label set may destroy it.

**Theorem 5.3** (Disambiguation Paradox)**.** *There exists a partial concept class $\mathbb{H} \subseteq (\mathcal{Y} \cup \{\star\})^{\mathcal{X}}$ such that:*

- $\mathrm{DS}\,(\mathbb{H}) = 1$, *and hence $\mathbb{H}$ is realizably PAC learnable.*

- *For every disambiguation scheme $\varphi_{\mathrm{bi}}$ with $|\mathcal{Y}_{\varphi_{\mathrm{bi}}}| = 2$, it satisfies that $\mathrm{DS}\,(\bar{\mathbb{H}}_{\varphi_{\mathrm{bi}}}) = \infty$.*

- *There exists a disambiguation scheme $\varphi_{\mathrm{mul}}$ such that $|\mathcal{Y}_{\varphi_{\mathrm{mul}}}| = \infty$, while $\mathrm{DS}\,(\bar{\mathbb{H}}_{\varphi_{\mathrm{mul}}}) = 1$.*

The idea of the construction is presented in Figure 1. The proof is presented in Appendix E.

As shown in Figure 1, the key point is that DS shattering is witnessed by pseudo-cubes, which requires the class to admit neighbors *independently* across coordinates. When a disambiguation reuses only a small set of labels, e.g., binary disambiguation, many distinct off-support behaviors are forced to collide on either 0 or 1, which can inadvertently create the combinatorial flexibility needed to realize large shattering.

In contrast, $\varphi_{\mathrm{mul}}$ assigns a hypothesis-specific symbol $\perp_h$ off-support; whenever $\perp_h$ appears on a coordinate it effectively identifies the underlying hypothesis, globally coupling all coordinates and blocking the neighbor structure needed for high-dimensional pseudo-cubes. Indeed, if a pseudo-cube of dimension $> 1$ existed and $h(x) = \perp_g$ for some

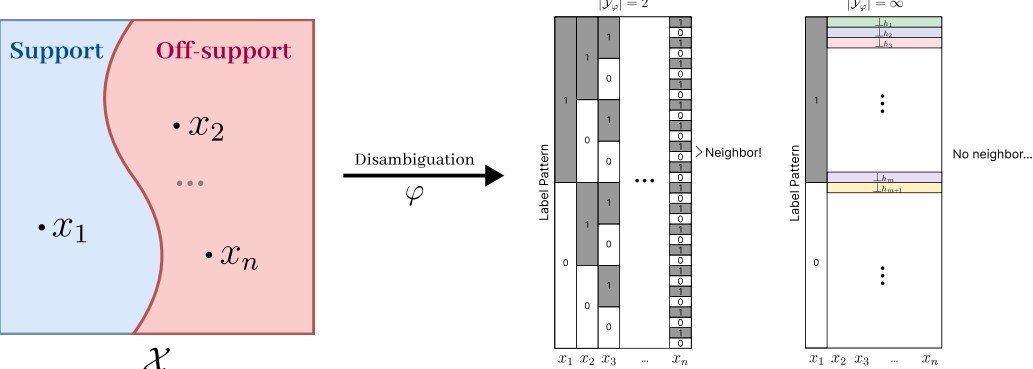

**Figure 1.** Overall idea of Theorem 5.3: There exists a partial concept class $\mathbb{H}$ for which any disambiguation scheme using only labels $\{0, 1\}$ causes the DS-dimension to blow up, whereas disambiguation with multiple labels may preserve the DS-dimension.

$x \in S$, where $S$ witness the shattering, then the pseudo-cube condition would require a neighbor $h'$ that agrees with $h$ on $S \setminus \{x'\}$ for some $x' \neq x$, and hence also outputs $\perp_g$ at $x$, which is impossible since $\perp_g$ is private to $g$.

At a high level, the hypothesis-specific label serves as an information channel that reveals the identity of the underlying hypothesis to the learner, explaining why a larger off-support label budget can instead preserve DS-dimension and hence learnability. However, as illustrated in Figure 1, the construction of $\varphi_{\text{mul}}$ clearly uses more disambiguation labels than necessary. From an information-theoretic perspective, one might hope to preserve the DS-dimension using far fewer labels by exploiting entropy more efficiently. This leads to a natural question: for a partial concept class whose expressive power is amplified by disambiguation, what is the minimal number of additional labels needed to keep the DS-dimension bounded? We investigate this question in next section.

### 5.2. Quantifying Disambiguation Label Complexity

The main object of study in this section is a family of partial concept classes $\mathbb{H}_n$ introduced by Alon et al. (2021b). In fact, the partial class $\mathbb{H}$ that exhibits the disambiguation paradox is defined as the limit of $\mathbb{H}_n$ as $n \to \infty$, hence $\mathbb{H}_n$ could be viewed as finite but still highly expressive versions of $\mathbb{H}$. We now quantify the minimal number of disambiguation labels required to ensure that the DS-dimension of $\mathbb{H}$ remains below a given threshold $d$, formalized by the following notion:

**Definition 5.4** (Disambiguation number). Let $\mathbb{H} \subseteq (\mathcal{Y} \cup \{\star\})^{\mathcal{X}}$ be a partial concept class with finite DS-dimension. For $d \geq \text{DS}(\mathbb{H})$, the disambiguation number $\mathfrak{D}_d(\mathbb{H})$ is the minimum number of off-support labels required by a disambiguation scheme whose induced total class has DS-

dimension at most $d$:

$$\mathfrak{D}_d(\mathbb{H}) := \min_{\varphi:\text{DS}(\mathbb{H}_\varphi) \leq d} |\mathcal{Y}_\varphi|.$$

The following theorem gives a quantitative relationship between the number of disambiguation labels and the resulting DS-dimension of $\mathbb{H}_n$, formalizing the extent to which efficiently exploiting label entropy can control the DS-dimension.

**Theorem 5.5.** *The disambiguation number of partial concept classes* $\mathbb{H}_n \subseteq (\mathcal{Y} \cup \{\star\})^{[n]}$ *satisfy that*

$$\mathfrak{D}_d(\mathbb{H}_n) = n^{\Omega\left(\frac{\log n}{d}\right)},$$

*or equivalently, any disambiguation with $k$ labels must have DS-dimension*

$$d \geq \Omega\left(\frac{\log^2 n}{\log n + \log k}\right).$$

The proof involves the fact that $|\mathbb{H}_n| \geq n^{\Omega(\log n)}$ (Alon et al., 2021b) and the generalized Sauer-Shelah lemma for Natarajan dimension (Natarajan, 1989; Haussler & Long, 1995; Daniely & Shalev-Shwartz, 2014). We leave the detail to Appendix F.

Since the DS-dimension equivalently describes sample complexity, this theorem quantitatively illustrates the trade-off between sample complexity and label complexity: in certain learning tasks, maintaining low sample complexity inevitably requires the introduction of additional labels to disambiguate the hypothesis class. In other words, label expansion is not merely a representational convenience, but a fundamental resource for learnability, highlights disambiguation as a complexity–control mechanism in the multiclass partial learning setting.

# 6. Conclusion

In this work, we study the PAC learnability of multiclass partial concept classes with possibly infinite label spaces, and showed that the DS-dimension characterizes learnability in this setting. We also study disambiguation schemes that complete hypotheses off-support, showing that the off-support label budget can substantially affect the induced DS-dimension, and quantified this effect through lower bounds on the labels required to preserve a target DS complexity. An interesting direction is to identify structural conditions under which one can design computationally efficient disambiguations that control the induced DS-dimension with a small off-support label budget.

# Acknowledgements

This work is supported by the Key R&D Program of Hubei Province under Grant 2024BAB038 and the National Key R&D Program of China under Grant 2023YFC3604702.

# Impact Statement

This paper presents work whose goal is to advance the field of Machine Learning. There are many potential societal consequences of our work, none which we feel must be specifically highlighted here."

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

# A. The One-inclusion Graph and Associated Proofs

## A.1. The One-inclusion Graph Algorithm

The one-inclusion graph (Haussler et al., 1994), is defined as follows.

**Definition A.1** (One-inclusion Graph). The *one-inclusion* graph of $\mathcal{H} \subseteq \mathcal{Y}^n$ is a hypergraph $\mathcal{G}(\mathcal{H}) = (V, E)$ that is defined as follows. The vertex-set is $V = \mathcal{H}$. For each $i \in [n]$ and $f : [n] \setminus \{i\} \to \mathcal{Y}$, let $e_{i,f}$ be the set of all $h \in \mathcal{H}$ that agree with $f$ on $[n] \setminus \{i\}$. The edge-set is

$$E = \big\{ (e_{i,f}, i) : i \in [n], f : [n] \setminus \{i\} \to \mathcal{Y}, e_{i,f} \neq \emptyset \big\}. \tag{2}$$

We say that the edge $(e_{i,f}, i) \in E$ contains the vertex $v$, and write $v \in (e_{i,f}, i)$, if $v \in e_{i,f}$. The size of the edge $(e_{i,f}, i)$ is defined to be $|(e_{i,f}, i)| := |e_{i,f}|$. We shall refer to hyperedges $e_{i,\mu} \in E$ with $|e_{i,\mu}| = 1$ as singletons.

Before formally introduce the one-inclusion graph algorithm, we define the *orientation* of a hypergraph and its *maximum out-degree*.

**Definition A.2** (Orientation). An orientation of the hypergraph $(V, E)$ is a mapping $\sigma : E \to V$ such that $\sigma(e) \in e$ for each $e \in E$. The out-degree of $v \in V$ in $\sigma$ is

$$\mathbf{outdeg}(v; \sigma) = |\{e : v \in e \wedge \sigma(e) \neq v\}|.$$

The maximum out-degree of $\sigma$ is

$$\mathbf{outdeg}(\sigma) = \sup_{v \in V} \mathbf{outdeg}(v; \sigma).$$

Then the following is the one-inclusion graph algorithm for a partial concept class $\mathbb{H}$, denoted as $\mathcal{A}^{\mathrm{OIG}}$.

---

**Algorithm 2** The one-inclusion algorithm $\mathcal{A}_{\mathbb{H}}^{\mathrm{OIG}}$ for partial concept class $\mathbb{H} \subseteq (\mathcal{Y} \cup \{\star\})^{\mathcal{X}}$

**Input:** An $\mathbb{H}$-realizable sample $S = \{(x_1, y_1), \ldots, (x_n, y_n)\}$.
**Output:** A hypothesis $\mathcal{A}_{\mathbb{H}}^{\mathrm{OIG}}(S) = h_S : \mathcal{X} \to \mathcal{Y}$.

For each $x \in \mathcal{X}$, the value $h_S(x)$ is computed as follows.

1: Define $\mathcal{H}$ as the class of all total functions $h : \{x_1, \ldots, x_n, x\} \to \mathcal{Y}$ s.t. $\{(z, h(z)) : z \in \{x_1, \ldots, x_n, x\}\}$ is realizable w.r.t $\mathbb{H}$.
2: Define the class of all label patterns over $\mathcal{H}$ as $\mathcal{H}' \subseteq \mathcal{Y}^{n+1}$.
3: Find an orientation $\sigma$ of $\mathcal{G}(\mathcal{H}')$ that minimizes the maximum out-degree.
4: Consider the edge in direction $n + 1$ defined by $S$; let

$$e = \{h \in \mathcal{H} : \forall i \in [n] \ \ h(x_i) = y_i\}.$$

5: Let $h' = \sigma((e, n + 1))$.
6: Set $h_S(x) = h'(n + 1)$.

---

We cite some lemmas that are useful in our proofs:

**Lemma A.3** (Lemma 13 of (Brukhim et al., 2022)). *If $\mathcal{H} \subseteq \mathcal{Y}^{d+1}$ has DS-dimension $d$, then there exists an orientation $\sigma$ of $\mathcal{G}(\mathcal{H})$ with $\mathbf{outdeg}(\sigma) \leq d$.*

**Lemma A.4** (Lemma 17 of (Brukhim et al., 2022)). *Let $\mathcal{H} \subseteq [p]^n$ be a class with Natarajan dimension $d_N < \infty$. Then there exists an orientation $\sigma$ of $\mathcal{G}(\mathcal{H})$ with maximum out-degree*

$$\mathbf{outdeg}(\sigma) \leq 20 d_N \ln(p).$$

## A.2. Theoretical Guarantee for the One-inclusion Graph Algorithm $\mathcal{A}_{\mathbb{H}}^{\mathrm{OIG}}$

*Proof of Theorem 4.2* Let $\mathcal{H}$ be the total class as defined in Algorithm 2, and $y' = (y'_1, \ldots, y'_{d+1})$ be a vertex in $\mathcal{G}(\mathcal{H})$. Let $\sigma$ be the orientation that minimizes the maximum out-degree of $\mathcal{G}(\mathcal{H})$ chosen by $\mathcal{A}_{\mathbb{H}}$, then by Lemma A.3, the maximum out-degree of $\sigma$ is at most $d$. Let $e_i$ be the edge in the $i$'th direction containing $y'$. For every $i \in [d + 1]$, we have $h_{S'_{-i}}(x'_i) \neq y'_i$

if and only if $\sigma(e_i) \neq y'$. Hence

$$\sum_{i=1}^{d+1} \mathbb{1}\left[h_{S'_{-i}}(x'_i) \neq y'_i\right] = \sum_{i=1}^{d+1} \mathbb{1}\left[\sigma(e_i) \neq y'\right] = \mathbf{outdeg}(y'; \sigma) \leq d.$$

It follows that there exists $i$ such that $h_{S'_{-i}}(x'_i) = y'_i$ □

### A.3. Learning a Menu for $\mathbb{H}$

*Proof of Theorem 4.5* The argument follows closely a proof of an analogous result by (Brukhim et al., 2022) for total concept class learning. Let $d = d_{DS}$. Define mapping $\mu_S : \mathcal{X} \to 2^{\mathcal{Y}}$:

$$\mu_S(x) = \{\mathcal{A}_{\mathbb{H}}^{\mathrm{OIG}}(T)(x) | T \subseteq S, |T| = d\}. \tag{3}$$

For a given $\mathbb{H}$-realizable sample $S$. Let $S_1$ be a subsample of $S$ with size $d + t$, chosen according to $S_1 \sim U_S^{d+t}$, where $U_S$ denote the uniform distribution over the $n$ examples in $S$. Denote $\mu_{S_1}$ by $\mu_1$. By definition, $\mu_1$ has size $\binom{d+t}{d}$. We then want to lower bound the number of examples in $S$ that is correctly labeled by $\mu_1$.

By the leave-one-out argument (Haussler et al., 1994), we have

$$\Pr_{(S_1,(x,y))\sim U_S^{d+t+1}}[y \in \mu_1(x)] = \Pr_{(S',I)\sim U_S^{d+t+1}\times U([d+t+1])}\left[y'_I \in \mu_{S'\setminus\{(x'_I,y'_I)\}}(x'_I)\right],$$

where $(x'_I, y'_I)$ is the $I$-*th* example in $S'$. The right-hand side is at least $\alpha = \frac{t+1}{d+t+1}$; otherwise there exists $T' \subseteq S'$ with $|T'| = d + 1$ such that for all $(x, y) \in T'$ we have $y \notin \mu_{S'\setminus\{(x,y)\}}(x)$. Since $T'\setminus\{(x,y)\} \subseteq S'\setminus\{(x,y)\}$, by (3) this would imply $\mathcal{A}_{\mathbb{H}}(T'\setminus\{(x,y)\})(x) \neq y$ for all $(x,y) \in T'$, contradicting Theorem 4.2. Therefore, in expectation at least an $\alpha$-fraction of examples $(x_i, y_i)$ in $S$ satisfy $y_i \in \mu_1(x_i)$. Remove all such examples from $S$ and repeat the argument on the remaining sample. At each step $j$ we obtain a subsample $S_j$ and a menu $\mu_j = \mu_{S_j}$ that covers at least an $\alpha$-fraction of the remaining examples. After $\ell$ steps, all examples are covered as soon as $(1 - \alpha)^\ell n < 1$. A sufficient choice is $\ell = \lfloor \frac{d+t+1}{t+1} \log(n) \rfloor$. Let $\tilde{S} \subseteq S$ be the concatenation of $S_1, \ldots, S_\ell$, and define the reconstruction function by

$$\rho_1(\tilde{S})(x) = \bigcup_{j=1}^{\ell} \mu_j(x) \triangleq \mu(x).$$

Then $\mu$ has size $p \leq l \times \binom{d+t}{d} \leq \binom{d+t+1}{t+1} \log(n)$, and $r_1 \leq l(d+t) \leq \frac{d+t+1}{t+1}(d+t)\log(n)$. □

### A.4. Natarajan Classes with Menu Realizability are Learnable

*Proof of Lemma 4.7* Let $\mathcal{D}$ be a distribution that is realizable by $\mathbb{H}$ and $\mu$. By the symmetrization argument,

$$\Pr_{(S,(x,y))\sim\mathcal{D}^{n+1}}[h_S(x) \neq y] = \Pr_{(S',I)\sim\mathcal{D}^{n+1}\times U([n+1])}\left[h_{S'_{-I}}(x'_I) \neq y'_I\right],$$

where $h_S$ is the output of Algorithm 1 by giving $S$, i.e., $h_S = \mathcal{A}_{\mathbb{H},\mu}^{\mathrm{OIG}}(S)$. Define $\mathcal{H}'$ be the same as that in the algorithm, then the Natarajan dimension of $\mathcal{H}'$ satisfies $\mathrm{Ndim}(\mathcal{H}') \leq \mathrm{Ndim}(\mathbb{H}) = d_N$. Denote by $\sigma$ the orientation of $\mathcal{G}(\mathcal{H}')$ that the algorithm chooses and $y'$ the vertex in $\mathcal{G}(\mathcal{H}')$ defined by $y'_1, \ldots, y'_{n+1}$, then by the proposition that the success probability of the algorithm is determined by the out-degrees of the orientation, we have

$$\Pr_{(S',I)\sim\mathcal{D}^{n+1}\times U([n+1])}\left[h_{S'_{-I}}(x'_I) \neq y'_I\right] = \frac{\mathbf{outdeg}(y';\sigma)}{n+1} \leq \frac{20d_N \ln(p)}{n+1},$$

where the inequality is derived by utilizing Lemma A.4, and we complete the proof. □

*Proof of Theorem 4.9* Given a sample $S$ that is realizable by both $\mathbb{H}$ and $\mu$, for each $T \subseteq S$ with size $m$, define $h_T = \mathcal{A}_{\mathbb{H},\mu}^{\mathrm{OIG}}(T)$ to be the hypothesis outputted by Algorithm 1 on input sample $T$. Let $\mathcal{H}_m = \{h_T : T \subseteq S, |T| = m\}$.

Let $m = \lceil 80d_N \log(p) \rceil$. Then for every distribution $\mathcal{Q}$ on $\mathcal{X} \times \mathcal{Y}$, Lemma 4.7 implies that

$$\Pr_{(T,(x,y))\sim\mathcal{Q}^{m+1}}[h_T(x) \neq y] \leq \frac{20d_N \log(p)}{m} \leq \frac{1}{4}.$$

By the (von Neumann, 1928)'s minimax theorem, there exists a distribution $\mathcal{P}$ on $\mathcal{H}_m$ such that

$$\Pr_{h \sim \mathcal{P}} [h(x) \neq y] \leq \frac{1}{4}.$$

By the definition of $\mathcal{H}_m$, we can view $\mathcal{P}$ as a distribution over samples with size $m$. Let $S_1, \ldots, S_l$ be i.i.d. samples drawn from $\mathcal{P}$. For each $(x, y) \in S$, by the concentration of measure,

$$\Pr\left[\frac{1}{\ell} \sum_{j=1}^{\ell} \mathbb{1}(h_{S_j}(x) \neq y) \geq \frac{1}{2}\right] \leq \exp\left(-\frac{\ell}{8}\right).$$

Setting $l = \lfloor 8 \log(n) \rfloor$, the right hand side of the equation above is strictly less than $\frac{1}{n}$. Then by union bound, with positive probability, for every $(x, y) \in S$ we have $\frac{1}{\ell} \sum_{j=1}^{\ell} \mathbb{1}_{h_{S'_j}(x) \neq y} < \frac{1}{2}$. This implies that there exists $S_1, \ldots, S_l$ such that the plurality vote over $h_{S_1}, \ldots, h_{S_l}$ correctly classifies all of $S$. The concatenation is the required $S'$. Finally we have $r_2 \leq |S'| = lm \leq 640 d_N \log(p) \log(n)$.  □

### A.5. Final Compression Scheme

*Proof of Theorem 4.10.* Let $S$ be an $\mathbb{H}$-realizable sample of size $n$. By Theorem 4.5, there is an $n \rightarrow r_1$ list sample compression scheme $(\kappa_1, \rho_1)$ for $\mathbb{H}$. Denote the outputted $p$-menu as $\mu$, $S$ is then $\mu$-realizable. By Theorem 4.9, there is an $n \rightarrow r_2$ sample compression scheme for $\mathbb{H}$ and $\mu$. The composition of the two schemes is an $n \rightarrow r_1 + r_2$ sample compression scheme for $\mathbb{H}$.  □

## B. Natarajan Dimension Fails to Characterize Multiclass Partial Learnability

*Proof of Theorem 3.1.* The construction is based on the pseudo-cube classes introduced by Brukhim et al. (2022). By their Theorem 2, for every integer $d \geq 1$, there exists a multiclass hypothesis class

$$\mathcal{H}_d \subseteq \mathcal{Y}_d^{\mathcal{X}_d}, \qquad |\mathcal{X}_d| = d,$$

such that $\mathrm{Ndim}(\mathcal{H}_d) = 1$ while $\mathcal{H}_d$ contains a $d$-dimensional pseudo-cube, and hence its DS-dimension is at least $d$. Assume w.l.o.g. that the domains and label sets are pairwise disjoint:

$$\mathcal{X}_d \cap \mathcal{X}_{d'} = \emptyset, \qquad \mathcal{Y}_d \cap \mathcal{Y}_{d'} = \emptyset \quad \text{for } \forall d \neq d'.$$

Define $\mathcal{X} = \cup_{d \geq 1} \mathcal{X}_d, \mathcal{Y} = \cup_{d \geq 1} \mathcal{Y}_d$. For each $d \geq 1$ and $h \in \mathcal{H}_d$, define a partial concept $\varphi_{d,h} : \mathcal{X} \rightarrow \mathcal{Y} \cup \{\star\}$ :

$$\varphi_{d,h}(x) = \begin{cases} h(x), & x \in \mathcal{X}_d, \\ \star, & x \notin \mathcal{X}_d. \end{cases}$$

Let $\mathbb{H} = \{\varphi_{d,h} : d \geq 1, h \in \mathcal{H}_d\}$. We have $\mathrm{Ndim}(\mathbb{H}) \geq 1$ since each $\mathcal{H}_d$ contains at least two hypotheses that disagree on some point of $\mathcal{H}_d$, the class $\mathbb{H}$ can realize two distinct labels on at least one point. Now we prove that $\mathrm{Ndim}(\mathbb{H}) \leq 1$.

Let $x, x' \in X$ be two distinct points. There are two cases to discuss.

(i). If $x \in \mathcal{X}_d$ and $x' \in \mathcal{X}_{d'}$ with $d \neq d'$, then for every hypothesis $\varphi_{d,h} \in \mathbb{H}$ we have $\varphi_{d,h}(x') = \star$. Consequently, $\mathbb{H}$ cannot realize the full $2 \times 2$ label pattern required for Natarajan shattering on $\{x, x'\}$.

(ii). If $x, x' \in \mathcal{X}_d$ for the same $d$, $\mathbb{H}|_{\{x,x'\}}$ coincides with the restriction of $\mathcal{H}_d|_{\{x,x'\}}$. Since $\mathrm{Ndim}(\mathcal{H}_d) = 1$, the set $\{x, x'\}$ cannot be Natarajan-shattered.

Combining with the lower bound yields $d_N(\mathbb{H}) = 1$.

It remains to show that $\mathbb{H}$ is not PAC learnable in the realizable setting. For every fixed $d > 0$, $\mathbb{H}$ contains a $d$-dimensional pseudo-cube, therefore has DS-dimension at least $d$. As $d$ could be arbitrarily large, it is not PAC learnable by Theorem 2.10.  □

## C. Upper Bounds of Sample Complexity for partial DS-classes

We first cite the well-known compression bounds for robust loss:

**Lemma C.1** (David et al., 2016). *Let $\mathcal{H} \subseteq \mathcal{Y}^{\mathcal{X}}$ be a hypothesis class with an $n \to r$ sample compression scheme $(\kappa, \rho)$. Then there exist learning algorithms $\mathcal{A}^{re}$ and $\mathcal{A}^{ag}$ s.t. for every distribution $\mathcal{D}$ on $(\mathcal{X} \times \mathcal{Y})$, with probability at least $1 - \delta$ over the choice of $S \sim \mathcal{D}^m$ with $m \geq 2r$,*

$$R_{\mathcal{D}}(\mathcal{A}^{re}(S)) \leq O\left(\frac{r \log(n) + \log(1/\delta)}{n}\right)$$

$$R_{\mathcal{D}}(\mathcal{A}^{ag}(S)) \leq \inf_{h \in \mathcal{H}} R_{\mathcal{D}}(h) + O\left(\sqrt{\frac{r \log(n) + \log(1/\delta)}{n}}\right).$$

*Proof of Theorem 4.11.* Let $d = \text{DS}(\mathbb{H})$. By Theorem 4.10, for every $t > 0$, there is an $n \to r$ sample compression scheme for $\mathbb{H}$ where

$$r \leq \left(\frac{d+t+1}{t+1}(d+t) + 640 \text{Ndim}(\mathbb{H}) \log\left(\binom{d+t+1}{t+1} \log(n)\right)\right) \log(n).$$

Let $t = \lceil d^{1/2} \rceil$, note that $\text{Ndim}(\mathbb{H}) \leq \text{DS}(\mathbb{H}) = d$, we have $r \leq O(d^{\frac{3}{2}} \log^2 n)$. Thus invoking Lemma C.1, there exists algorithms $\mathcal{A}^{re}$ and $\mathcal{A}^{ag}$ such that

$$R_{\mathcal{D}}(\mathcal{A}^{re}(S)) \leq O\left(\frac{d^{\frac{3}{2}} \log^3 n + \ln(1/\delta)}{n}\right),$$

and

$$R_{\mathcal{D}}(\mathcal{A}^{ag}(S)) \leq \inf_{h \in \mathcal{H}} R_{\mathcal{D}}(h) + O\left(\sqrt{\frac{d^{\frac{3}{2}} \log^3 n + \ln(1/\delta)}{n}}\right).$$

Setting these less than $\varepsilon$ and solving for sufficient sizes of $m$ to achieve these yields sample complexity bounds in realizable case and agnostic case, respectively. $\square$

## D. Lower Bounds of Sample Complexity for partial DS-classes

We first present a technical lemma:

**Lemma D.1.** *Let $X$ be a $(m, p)$ binomial variable and assume that $p = \frac{1-\varepsilon}{2}$. Then*

$$\Pr[X \geq m/2] \geq \frac{1}{2}\left(1 - \sqrt{1 - e^{\frac{-m\varepsilon^2}{1-\varepsilon^2}}}\right).$$

*Proof of Theorem 4.12.* Specifically, for any $d \leq \text{DS}(\mathbb{H})$, let $\mathcal{X}_d\{x_1, \ldots, x_d\}$ be a set DS-shattered by $\mathbb{H}$, and let $\mathbb{H}_d$ be the class of all total (non-partial) functions $\mathcal{X}_d \to \mathcal{Y}$. Any distribution $P$ on $\mathcal{X}_d \times \mathcal{Y}$ w.r.t. $\mathbb{H}_d$ can be extended to a distribution on $\mathcal{X} \times \mathcal{Y}$ w.r.t. $\mathbb{H}$ with $\Pr((\mathcal{X} \backslash \mathcal{X}_d) \times \mathcal{Y}) = 0$. Thus, any lower bound on the sample complexity of PAC learning the total concept class $\mathbb{H}_k$ is also a lower bound on the sample complexity of learning $\mathbb{H}$.

For the realizable case, the proof follows (Daniely et al., 2011). Let $d = \text{DS}(\mathbb{H})$. Then there exists $S_{\mathcal{X}} = (x_1, \ldots, x_d)$ such that $\mathbb{H}|_{S_{\mathcal{X}}}$ contains a pseudo-cube $\mathcal{H}'$, i.e., for every $h \in \mathcal{H}'$ and $i \in [d]$ there exists $h' \in \mathcal{H}'$ such that $h(x_j) = h'(x_j)$ for all $j \neq i$ and $h(x_i) \neq h'(x_i)$. In particular, $\mathcal{M}_{re}^{\mathbb{H}} \geq \mathcal{M}_{re}^{\mathcal{H}'}$. Uniformly choose $h^*$ from $\mathcal{H}'$, and consider the marginal distribution $\mathcal{D}_{\mathcal{X}}$ of $\mathcal{D}$ defined by

$$\Pr(x_1) = 1 - 2\varepsilon, \qquad \Pr(x_i) = \frac{2\varepsilon}{d-1}(\forall i = 2, \ldots, d).$$

Consider the sample $S = \{(x_1, h^*(x_1)), \ldots, (x_m, h^*(x_m))\} \sim \mathcal{D}^m$ with $m \leq \frac{d-1}{6\varepsilon}$. Then by Chernoff's bound, with probability $\frac{1}{100}$, $S$ contains at most $\frac{d-1}{2}$ examples from $S \backslash \{x_1\}$. Call this event by $E$. Conditioned on $E$, the distribution of the unobserved points is uniform among the vertices of the $d/2$-dimensional pseudo-cube. Thus if the test point falls

in $S \setminus \{x_1\}$, any learner will make wrong prediction with probability at least $1/2$, since every hyperedge has size at least 2 and once the hyperedge(s) is determined, all vertexes contained in the hyperedge are equally likely to be chosen as the prediction. Because $\Pr(E) \geq 1/100$, we can see that $\mathcal{M}_{re}(\varepsilon, \delta) = \Omega(\frac{d}{\varepsilon})$. For the other part of the bound, notices that the probability that $S$ includes only $x_1$ is $(1 - 2\varepsilon)^m \geq e^{-4\varepsilon m}$, which is more than $\delta$ if $m \leq \frac{\ln(1/\delta)}{4\varepsilon}$. We there for obtain that

$$\mathcal{M}_{re}(\varepsilon, \delta) \geq \max\left\{ \frac{d-1}{6\varepsilon}, \frac{\ln(1/\delta)}{2\varepsilon} \right\} \geq C_1 \left( \frac{\mathrm{DS}\,(\mathbb{H})}{\varepsilon} + \frac{\log(1/\delta)}{\varepsilon} \right)$$

for some constant $C_1 > 0$.

The lower bound for the agnostic case follows immediately by $\mathcal{M}_{ag}(\varepsilon, \delta) \geq \mathcal{M}_{re}(\varepsilon, \delta)$. $\qquad\square$

# E. The Construction of Disambiguation Paradox

*Proof of Theorem 5.3.* Consider the partial concept class $\mathbb{H}$ from Theorem 5.1. It remains to prove the last item by constructing a disambiguation $\bar{\mathbb{H}}_{\mathrm{mul}}$ that preserves DS-dimension.

Since $\bar{\mathbb{H}}$ is countable, for each $h \in \mathbb{H}$, assign a unique $\perp_h \in \mathcal{Y}$ to $h$ and consider $\bar{h} \subseteq \mathcal{Y}^{\mathcal{X}}$:

$$\bar{h}(x) = \begin{cases} h(x), & x \in \mathrm{supp}(h), \\ \perp_h, & \text{otherwise.} \end{cases}$$

Then $\bar{\mathbb{H}}_{\mathrm{mul}} = \{\bar{h} : h \in \mathbb{H}\}$ is a disambiguation of $\mathbb{H}$ where the labels $\perp_h$ are all distinct. We now show that $\mathrm{DS}\,(\bar{\mathbb{H}}_{\mathrm{mul}}) = 1$.

First, $\mathrm{DS}\,(\bar{\mathbb{H}}_{\mathrm{mul}}) \geq 1$ since $\mathrm{DS}\,(\mathbb{H}) = 1$ implies that there exists a point $x \in \mathcal{X}$ such that $\mathbb{H}|_{\{x\}}$ contains at least two distinct labels. As $\bar{h}$ extends $h$ on $\mathrm{supp}(h)$, the restriction $\bar{\mathbb{H}}_{\mathrm{mul}}|_{\{x\}}$ also contains at least two distinct labels, and hence contains a 1-dimensional pseudo-cube.

Next we prove $\mathrm{DS}\,(\bar{\mathbb{H}}_{\mathrm{mul}}) \leq 1$. Fix any two distinct points $a \neq b$ and let $F = \{a, b\}$. Suppose toward a contradiction that $\bar{\mathbb{H}}_{\mathrm{mul}}|_F$ contains a 2-dimensional pseudo-cube $C$.

Take any $u \in C$ and write $u = \bar{h}|_F$ for some $h \in \mathbb{H}$. Assume w.l.o.g $u(a) = \perp_h$, then apply the pseudo-cube property to $u$ at coordinate $b$: there must exist $w \in C$ such that $w(a) = u(a) = \perp_h$ and $w(b) \neq u(b)$. However, $\perp_h$ is a private label: the only hypothesis in $\bar{\mathbb{H}}_{\mathrm{mul}}$ that ever outputs $\perp_h$ is $\bar{h}$ itself. Hence $w = u$, contradicting $w(b) \neq u(b)$. This yields $\mathrm{DS}\,(\bar{\mathbb{H}}_{\mathrm{mul}}) \leq 1$.

Combining the two inequalities yields $\mathrm{DS}\,(\bar{\mathbb{H}}_{\mathrm{mul}}) = 1$, completing the proof.

# F. Disambiguation Number of $\mathbb{H}_n$

*Proof of Theorem 5.5.* Let $\mathbb{H}_n$ be the partial concept class from Theorem 11 of (Alon et al., 2021b), satisfying $|\mathbb{H}_n| \geq n^{\Omega(\log n)}$. Let $\bar{\mathbb{H}}_\varphi$ be a disambiguation using $k$ labels. Injectivity implies $|\bar{\mathbb{H}}_\varphi| = |\mathbb{H}_n|$.

We use the generalized Sauer-Shelah lemma for the Natarajan dimension (Natarajan, 1989; Haussler & Long, 1995; Daniely & Shalev-Shwartz, 2014). The cardinality is bounded by:

$$|\bar{\mathbb{H}}_\varphi| \leq \sum_{i=0}^{\mathrm{Ndim}\,(\mathbb{H})} \binom{n}{i} k^{2i} \leq \left(nk^2\right)^{\mathrm{Ndim}\,(\mathbb{H})}.$$

(Note: The factor $k^2$ or $k$ depends on the specific variant; asymptotically $k^{O(1)}$ suffices).

We now relate this to the DS-dimension. It is a standard result that $\mathrm{Ndim}\,(\mathbb{H}) \leq \mathrm{DS}\,(\mathbb{H})$ for any class $\mathbb{H}$ (Daniely & Shalev-Shwartz, 2014; Brukhim et al., 2022). Therefore, if we impose a constraint on the learnability such that $\mathrm{DS}\,(\bar{\mathbb{H}}_\varphi) \leq d$, we necessarily have $\mathrm{Ndim}\,(\bar{\mathbb{H}}_\varphi) \leq d$. Substituting this into the exponent of the Sauer bound (valid since the base $nk^2 \geq 2$):

$$|\bar{\mathbb{H}}_\varphi| \leq \left(nk^2\right)^d.$$

Combining with the lower bound on $|\mathbb{H}_n|$:

$$n^{\Omega(\log n)} \leq \left(nk^2\right)^d.$$

Taking logarithms:

$$\Omega(\log^2 n) \le d \left( \log n + 2 \log k \right).$$

The results then follow from straightforward algebraic manipulation. $\qquad\square$

