# OpenReview forum: "On Learnability and Disambiguation of Multiclass Partial Concept Classes"
_ICML.cc/2026/Conference — ICML 2026 regular_

### Official Review · Reviewer_JeTf · 2026-03-03

**Soundness:** 3
**Presentation:** 3
**Significance:** 3
**Originality:** 4
**Overall Recommendation:** 5
**Confidence:** 4

**Summary:**

This paper studies the PAC learnability of partial concept classes in the multiclass setting. The authors establish that the Daniely–Shalev dimension fully characterizes the learnability of multiclass partial concept classes, even when the label space is unbounded. In addition, they introduce the Disambiguation Paradox, showing that disambiguation schemes using fewer labels may actually increase the complexity of the induced class, whereas richer disambiguation schemes can preserve learnability.

**Compliance With Llm Reviewing Policy:**

Affirmed.

**Final Justification:**

All my concerns have been addressed.

**Key Questions For Authors:**

See the weaknesses.

**Strengths And Weaknesses:**

#Strengths
- The authors clearly explain why the existing theory of Kalavasis et al. (2022) does not extend to the case of unbounded label spaces, and they make a convincing case that a new characterization is needed. Extending the theory to the $|Y|=\infty$ setting is both natural and worthwhile.

- I particularly appreciate that the authors explicitly state which technical ingredients are inherited from prior work. This is very important in learning theory papers: it allows readers to quickly assess what is new versus what is adapted. This makes the manuscript reader-friendly and facilitates evaluation of originality.

- The disambiguation paradox reveals a counterintuitive phenomenon. Moreover, they provide a quantitative analysis of the resulting label–sample complexity trade-off.

- The sample compression used in the proof of the upper bounds mainly relies on previously developed tools. However, the formulation and analysis of the disambiguation paradox appear to be new.

I believe the paper makes a solid contribution to learning theory, particularly in clarifying the role of disambiguation and complexity in multiclass partial settings.

# Minor comments
- In the Introduction, Question 1 and Question 2 appear in different font styles.
- In Section 5, the class $\mathbb{H}_n$ is used but not explicitly defined. Although the authors state that it originates from Alon et al. (2021), providing an explicit definition (or at least a concise description) would improve readability.
- In the Appendix, the proof of Theorem 5.3 appears to be missing a concluding QED symbol.

---

> ### Author Rebuttal · Authors · 2026-03-29
>
> Thanks for the careful reading, we will correct the typos in our revision.

---

> > ### Author Rebuttal · Reviewer_JeTf · 2026-04-01
> >
> > I don't have major concerns and I support to accept this paper.

---

> > > ### Author Response · Authors · 2026-04-04
> > >
> > > Thanks for the comments.

---

### Official Review · Reviewer_zrKk · 2026-03-08

**Soundness:** 3
**Presentation:** 4
**Significance:** 2
**Originality:** 1
**Overall Recommendation:** 3
**Confidence:** 5

**Summary:**

The paper studies PAC learnability of multiclass partial concept classes with a label space that can potentially be infinite. The paper proves that a class is PAC learnable if and only if it has finite DS dimension. The authors also provide quantitative sample complexity bounds controlled by the same dimension. Finally, the authors show that using a small number of labels when disambiguating a class (filling undefined points with labels) may cause the DS dimension to blow-up.

**Compliance With Llm Reviewing Policy:**

Affirmed.

**Final Justification:**

In short, given that the paper is purely theoretical, I respectfully think it doesn't have sufficient technically novel/challenging contributions. Please see my rebuttal acknowledgment for more details.

**Key Questions For Authors:**

The lower bound for the agnostic setting stated in the paper is $\Omega(\frac{DS}{\epsilon^2})$ (ignoring the $\delta$ dependence). However, I couldn’t find such a lower bound for total classes. In fact a recent work by CEH+25$^1$ shows that the sample complexity of agnostic PAC learning of multiclass classes is $\frac{DS^{1.5}}{\epsilon}+\frac{Nat}{\epsilon^2}$ where the upper bound is tight up to a factor of $DS^{0.5}$. Does this contradict your lower bound? I think your argument on line 824 is incorrect and it only holds for binary classes. Say $h(x)=0$. It could be that in the sample we see labels $0, 0, 1, 2, 3, 4, 5, 6, 7, 8$ for $x$. In this case, even though more than half the labels differ from $h(x)$, the learner still predicts $0$.

Note: this potential issue did not affect my overall score.

1 Alon Cohen, Liad Erez, Steve Hanneke, Tomer Koren, Yishay Mansour, Shay Moran, Qian Zhang. Sample Complexity of Agnostic Multiclass Classification: Natarajan Dimension Strikes Back. 2025.

**Limitations:**

yes

**Strengths And Weaknesses:**

The paper is technically sound and the proofs seem correct for the most part with one potential issue raised below. The presentation is clear and the outline is easy to follow. However, I believe the contributions of the paper are limited as the main technical results seem to be a relatively straightforward extension of the results by AHHM21 and BCDMY22 (specifically, using compression and learning from menu). Thus, in my opinion the contributions are not enough for ICML.

---

> ### Author Rebuttal · Authors · 2026-03-29
>
> # About the lower bounds
>
> Thank you for pointing this out. We agree that the argument around line 824, as previously written, does not go through in the multiclass setting. In particular, that step implicitly invoked a binary majority-type reasoning, whereas in the multiclass case the learner is defined via plurality, and the two are not equivalent. The example in the comment correctly highlights this issue.
>
> Upon revisiting the proof, we found that this point is best addressed by adjusting the lower-bound argument at a more structural level. In the revised version, rather than deriving an agnostic lower bound of the form $\Omega(DS(H)/\epsilon^2)$ from this step, we directly invoke the realizable lower bound, which immediately yields $M_{\mathrm{ag}}(\epsilon,\delta)\ge \Omega\left(\frac{DS(H)}{\epsilon}\right),$
> since every realizable distribution is a special case of the agnostic setting.
>
> We believe this revision resolves the issue in a clean and principled way. Importantly, it does not affect our DS-dimension characterization or the main conclusions of the paper. We thank the reviewer again for drawing our attention to this point, which helped us improve both the correctness and the presentation of the lower-bound argument.
>
> # About the limitation of contribution
>
> Thank you for the thoughtful comment. We respectfully believe the contribution is not merely a straightforward extension of AHHM21 and BCDMY22. Our main contribution is conceptual: we show that the Natarajan dimension no longer characterizes learnability for multiclass partial concept classes when the label space is unbounded, and we identify the DS-dimension as the correct characterization in this regime. Thus, the paper does not only adapt existing techniques, but also pinpoints a genuine limitation of prior theory [1] and replaces it with the right notion.
>
> More importantly, we uncover the Disambiguation Paradox, which to the best of our knowledge has not been identified in prior work: a simpler disambiguation label space can destroy learnability, while a richer one can preserve it. We further quantify this through a label–sample complexity trade-off. Overall, the research investigates a core question, and the study's significant objective pertains to understanding both the correct complexity measure and the structural role of disambiguation in multiclass partial learning. We therefore view the paper’s contribution as primarily conceptual, not only technical.
>
> *Reference:*
>
> [1] Kalavasis, A., et al, *"A. Multiclass learnability beyond the PAC framework: Universal rates and partial concept classes"*. In NeurIPS, 2022.

---

> > ### Author Rebuttal · Reviewer_zrKk · 2026-04-02
> >
> > Thank you for your response. Unfortunately, in my opinion, the contributions are limited and I will keep my score.
> >
> > Regarding the right dimension: As total classes are a special case of partial classes, the results of BCDMY22 already implies that Natarjan dimension is not the correct dimension. Furthermore, it has already been noted that DS dimension is the right dimension for partial multiclass hypothesis classes BHM$^1$. Please see Definition 11 in BHM$^1$ and the remarks afterwards (specifically, the paragraph after Equation 6).
> >
> > 1. Nataly Brukhim, Steve Hanneke, Shay Moran. Improper Multiclass Boosting. COLT2023.

---

> > > ### Author Response · Authors · 2026-04-04
> > >
> > > Thank you for your thoughtful feedback. We would like to mildly note that, while BHM23 only claims the DS dimension is the “right” complexity measure, our work establishes formal upper and lower bounds in both the realizable and agnostic settings, providing the rigorous theoretical guarantees that were previously missing. Moreover, we uncover an unexpected disambiguation paradox, which highlights phenomena not addressed in prior work.

---

### Official Review · Reviewer_gn1c · 2026-03-08

**Soundness:** 4
**Presentation:** 4
**Significance:** 3
**Originality:** 3
**Overall Recommendation:** 5
**Confidence:** 4

**Summary:**

The paper provides upper an lower bound for learning multiclass partial classes. Prior work had a dependency on the number of classes which made the results vacuous for infinite label classes. Essentially the work uses the new characterization of multiclass learnability and applies these approaches in the partial label setting. Another result that the paper presents is a disambiguation paradox. We know that for learning binary partial concept classes disambiguation can break to learnability. For multiclass concept classes the paper shows that the choice of number of classes to disambiguate matters and may make the class learnable or not.

**Compliance With Llm Reviewing Policy:**

Affirmed.

**Final Justification:**

I maintain my evaluation as acceptance.

Strengths:
1) Soundness: The paper proves all assertions.
2) Significance: The paper bridges a clear gap in prior work.
3) Clarity: The paper is well-written and the results are clear.

Weaknesses:
1) Originality: The techniques used are drawn from recent progress in Sample Complexity of multiclass classification and adapted to this setting.

The rebuttal addressed my concerns and reinforced my assessment.

**Key Questions For Authors:**

1) Can you comment on what causes the discrepancy between 4.11 and 4.12?
2) The paper seems to borrow a lot of analysis and arguments from Brukhim et al. (2022), this is natural as it offered a lot of developments for characterizing multiclass learnability. Could you comment more on the difficulty to apply the approaches there for partial classes?

**Limitations:**

Yes

**Strengths And Weaknesses:**

Strengths:
1) Presentation: The paper seems well-written and easy to follow.
2) Soundness: The paper supports all the claims with rigorous proofs.
3) Significance: The paper provides a better characterization that prior work. Being close to provide fully tight upper and lower bounds.
4) Originality: The paper provides new understanding on this problem not present in the current literature.

Weaknesses:
1) The upper and the lower bounds differ polynomially and the paper does not comment on the what causes this difference.

---

> ### Author Rebuttal · Authors · 2026-03-27
>
> Thanks for your careful reading.
>
> # About the discrepancy between 4.11 and 4.12.
>
> Good quesion. In words, this gap is inherent to sample-compression-based methods. We in fact discuss this point explicitly in the paragraph “About the log(n) dependence” (Lines 309–323), where we explain why this discrepancy arises. For your convenience, we paste the relevant discussion here.
>
> Our constructed sample compression scheme has size scaling logarithmically with the sample size n. In the binary setting, it is a classical result that learnability implies the existence of a compression scheme whose size is independent of n (David et al., 2016). One might therefore wonder whether the log n factor in our construction is merely an artifact of the proof technique or instead reflects a fundamental feature of multiclass learning. A recent result of Pabbaraju (2024) resolves this question by showing that there exist multiclass concept classes with finite DS dimension that do not admit any sample compression scheme of size independent of n. This establishes an intrinsic separation between binary and multiclass learnability.
>
> # About the difficulty to apply the approaches
>
> The technical challenge is not in inventing entirely new machinery, but in carefully adapting and re-establishing these tools in the regime of multiclass partial concept classes, where one must build a PAC learning theory that also accounts for off-support behavior and disambiguation. In this sense, the contribution is not a black-box reuse of prior arguments, but a careful extension of the multiclass PAC framework to a substantially richer setting.

---

> > ### Author Rebuttal · Reviewer_gn1c · 2026-04-01
> >
> > The authors have addressed my questions.

---

> > > ### Author Response · Authors · 2026-04-04
> > >
> > > Thanks for the comments.

---

### Official Review · Reviewer_NQBU · 2026-03-09

**Soundness:** 4
**Presentation:** 2
**Significance:** 2
**Originality:** 2
**Overall Recommendation:** 3
**Confidence:** 5

**Summary:**

This paper studies PAC learning of partial concept classes in the multiclass setting with an arbitrary label space. It shows that the same combinatorial dimension (DS) that characterizes learnability in the multiclass case also characterizes the learnability of partial concepts, paralleling the binary setting where the VC dimension characterizes partial concept classes.

A central aspect of partial concepts is understanding how to complete them into total concept classes while minimizing the increase in the relevant combinatorial dimension. One of the main insights of this work is that richer label spaces make it possible to control this dimensional growth more effectively than small label spaces, an observation that is counterintuitive. This phenomenon reveals an interesting structural property of learning problems with large label spaces.

**Compliance With Llm Reviewing Policy:**

Affirmed.

**Key Questions For Authors:**

See my comments above.

**Strengths And Weaknesses:**

**Strengths**

Multiclass classification and partial concepts are both central frameworks in learning theory, making it natural to study their intersection more carefully. This paper completes the picture and reveals an interesting property of disambiguations of concept classes that depends on the size of the allowed label space, which in my view is the most interesting result of the paper.
The results rely on techniques from both frameworks and appear technically sound.

**Weaknesses**

The main weaknesses, described below, are somewhat limited technical novelty, an incomplete and somewhat misleading discussion of related work, and several instances of misleading phrasing.

*Technical novelty*

The theoretical results in Sections 3 and 4 rely largely on standard techniques from multiclass learning, and I did not identify a substantially new technical idea required to obtain them. The result in Section 5 is conceptually interesting, however, from a technical perspective it also appears to rely mostly on previously known ideas.

*Missing and misleading related work*

It seems that the authors missed several important related works that should be discussed more carefully.

- (l.218) "ERM has recently been shown to fail for binary partial concept classes and for multiclass total learning (Alon et al., 2021; Brukhim et al., 2022)": For multiclass learning, this phenomenon was already shown in earlier works, such as Multiclass Learnability and the ERM Principle (Best Student Paper, COLT'11) and Optimal Learners for Multiclass Problems (COLT'14).

Related work subsection:

- "The framework is then extended to the multiclass setting (Natarajan, 1989; Ben-David et al., 1995; Brukhim et al., 2022)": Again, the two papers mentioned above should be cited here as well.

- "regression (Simon, 1997)": Several crucial papers appear to be missing, including

  1. Scale-Sensitive Dimensions, Uniform Convergence, and Learnability (JACM'97)
  2. Prediction, Learning, Uniform Convergence, and Scale-Sensitive Dimensions (JCSS'98)
  3. Optimal Learners for Realizable Regression: PAC Learning and Online Learning (NeurIPS'23)

- "online learning (Littlestone, 1987; Ben-David et al., 2009)": Adversarial Laws of Large Numbers and Optimal Regret in Online Classification (STOC'21) should also be mentioned.

- "robust learning (Montasser et al., 2019; Xu & Liu, 2022; Montasser et al., 2022)": There are at least two papers that use partial concepts in this setup:
  1. Improved Generalization Bounds for Adversarially Robust Learning (JMLR'22), which shows a generalization bound for regression using partial concepts.
  2. A Characterization of Semi-Supervised Adversarially Robust PAC Learnability (NeurIPS'22), which shows an optimal semi-supervised sample complexity result using partial concepts.

When discussing partial concepts, another older paper could be mentioned: On Agnostic Learning with {0, ∗, 1}-Valued and Real-Valued Hypotheses (COLT'01).

Another potentially related paper is Comparative Learning: A Sample Complexity Theory for Two Hypothesis Classes (ITCS'23), which also relies on partial concepts.

**Other comments**

- (l.310, second column) “About the log(n) dependence”: In the form stated in the theorem (which concerns sample complexity, not a generalization rate), the dependence is actually not on $n$.

- (l.368) “there exists a learnable partial class for which every disambiguation yields a total class of unbounded VC-dimension and hence is not PAC learnable”: This statement is somewhat misleading. The partial concept class itself is learnable, even though after disambiguation the resulting total class may have unbounded VC-dimension and therefore is not PAC learnable.

---

> ### Author Rebuttal · Authors · 2026-03-27
>
> Thanks for the reviewing.
>
> # About the technical novelty
>
> Thank you for this thoughtful comment. We want to highlight that the results do not directly apply in the setting of multiclass partial concept classes with possibly unbounded label spaces, which is the main focus of the paper. The central contribution is to show that in this regime, PAC learnability is characterized by the DS-dimension, whereas the previously known Natarajan-based characterization only applies to finite label spaces [1]. This is precisely the gap our paper addresses. The technical challenge is not in inventing entirely new machinery, but in carefully adapting and re-establishing these tools in the regime of multiclass partial concept classes, where one must build a PAC learning theory that also accounts for off-support behavior and disambiguation. In this sense, the contribution is not a black-box reuse of prior arguments, but a careful extension of the multiclass PAC framework to a substantially richer setting.
>
> Regarding Section 5, we would like to highlight that, disambiguation is a central issue in learning partial concept classes, since any learner must effectively decide how to behave outside the support of the target concept. While this issue has been recognized in the binary partial setting, surprisingly, to the best of our knowledge it has not been systematically studied in the multiclass setting. The main intended contribution of this section is primarily conceptual: we identify a new phenomenon—the Disambiguation Paradox—showing that restricting to a smaller disambiguation label set can increase the induced DS-dimension and destroy learnability, whereas allowing richer off-support labels can preserve learnability. We further quantify this through a label-complexity/sample-complexity trade-off. To our knowledge, this phenomenon and this formulation **have not been identified previously** in the multiclass partial-learning literature.
>
> [1] Kalavasis, A., et al, *"A. Multiclass learnability beyond the PAC framework: Universal rates
> and partial concept classes"*. In NeurIPS, 2022.
>
> # About the missing and misleading related work
>
> We thank the reviewer for pointing these out. We will incorporate and discuss these related works in the revised version.

---

> > ### Author Rebuttal · Reviewer_NQBU · 2026-04-04
> >
> > Thank you for your response.
> > I still believe that the results in Sections 3 and 4 are largely a straightforward adaptation of total classes. The results in Section 5 are indeed interesting. However, I remain unconvinced that the overall contribution meets the bar for acceptance.
> >
> > In particular, the point raised by reviewer zrKk regarding the paper Improper Multiclass Boosting (COLT 2023)—specifically Definition 11 and the subsequent remarks—reinforces concerns about the novelty of the methods developed in Sections 3 and 4.

---

> > > ### Author Response · Authors · 2026-04-04
> > >
> > > Thank you for your thoughtful feedback. Please let us mildly note that while BHM23 only claims that the DS dimension is the “right” complexity measure, our work provides formal upper and lower bounds in both realizable and agnostic settings, establishing the rigorous theoretical guarantees that were previously missing.

---

### Decision · Program_Chairs · 2026-04-30

**Decision:**

Accept (regular)

**Comment:**

This paper induced a long discussion among reviewers which ended with an agreement on accepting this paper. But we strongly recommend the following points to be addressed for the camera ready version.

**Correction of the Lower Bound:** As Reviewer zrKk pointed out and acknowledged by the authors during the rebuttal, the original agnostic lower bound argument contained an error regarding multiclass plurality versus binary majority. The camera-ready version must formally reflect the corrected proof utilizing the realizable lower bound argument as agreed upon in the rebuttal. Furthermore, a brief discussion contextualizing the bounds against the recent $1/\epsilon$ and $1/\epsilon^2$ rates identified in CEH+25 should be included.

CEH+25: Sample Complexity of Agnostic Multiclass Classification: Natarajan Dimension Strikes Back. 2025.

**Comprehensive Literature Review:** Reviewer NQBU provided a detailed list of missing citations spanning multiclass learning, regression, scale-sensitive dimensions, and online learning (e.g., Multiclass Learnability and the ERM Principle, Optimal Learners for Multiclass Problems). Additionally, the phrasing around ERM failure must be adjusted to avoid being misleading, and the connection to the BHM23 remarks regarding the DS dimension must be explicitly cited. The authors must thoroughly update the related work section to ensure all prior technical foundations are accurately cited.

BHM23: Nataly Brukhim, Steve Hanneke, Shay Moran. Improper Multiclass Boosting. COLT2023.